# An Advanced Noise Reduction and Edge Enhancement Algorithm

**DOI:** 10.3390/s21165391

**Published:** 2021-08-10

**Authors:** Shih-Chia Huang, Quoc-Viet Hoang, Trung-Hieu Le, Yan-Tsung Peng, Ching-Chun Huang, Cheng Zhang, Benjamin C. M. Fung, Kai-Han Cheng, Sha-Wo Huang

**Affiliations:** 1Department of Electronic Engineering, National Taipei University of Technology, Taipei 10608, Taiwan; schuang@ntut.edu.tw; 2International Graduate Program in Electrical Engineering & Computer Science, National Taipei University of Technology, Taipei 10608, Taiwan; viethqict@gmail.com; 3Faculty of Information Technology, Hung Yen University of Technology and Education, Hungyen 17000, Vietnam; 4Department of Computer Science, National Chengchi University, Taipei 11605, Taiwan; 108753143@nccu.edu.tw (K.-H.C.); 108753138@nccu.edu.tw (S.-W.H.); 5Department of Computer Science, National Yang Ming Chiao Tung University, Hsinchu 30010, Taiwan; chingchun@cs.nctu.edu.tw; 6Department of Mechanical System Engineering, Ibaraki University, Ibaraki 318-0022, Japan; cheng.zhang.abbott@vc.ibaraki.ac.jp; 7School of Information Studies, McGill University, Montréal, QC H3A 1X1, Canada; ben.fung@mcgill.ca

**Keywords:** noise removal, deep image prior, edge enhancement, contrast enhancement

## Abstract

Complementary metal-oxide-semiconductor (CMOS) image sensors can cause noise in images collected or transmitted in unfavorable environments, especially low-illumination scenarios. Numerous approaches have been developed to solve the problem of image noise removal. However, producing natural and high-quality denoised images remains a crucial challenge. To meet this challenge, we introduce a novel approach for image denoising with the following three main contributions. First, we devise a deep image prior-based module that can produce a noise-reduced image as well as a contrast-enhanced denoised one from a noisy input image. Second, the produced images are passed through a proposed image fusion (IF) module based on Laplacian pyramid decomposition to combine them and prevent noise amplification and color shift. Finally, we introduce a progressive refinement (PR) module, which adopts the summed-area tables to take advantage of spatially correlated information for edge and image quality enhancement. Qualitative and quantitative evaluations demonstrate the efficiency, superiority, and robustness of our proposed method.

## 1. Introduction

Noise usually accompanies images during acquisition or transmission, resulting in contrast reduction, color shift, and poor visual quality. The interference of noise not only contaminates the naturalness of an image, but also damages the precision of various computer vision-based applications, such as semantic segmentation [1,2], motion tracking [3,4], action recognition [5,6], and object detection [7,8,9,10,11,12], to name a few. Consequently, noise removal for these applications has attracted great interest as a preprocessing task over the last two decades. The classical description for the additive noise model can be defined as follows [13]:(1)$$I_{n}\;=\;I_{gt}\;+\;\beta ,$$
where $I_{n}$, $I_{gt}$, and β represent an image accompanied with noise, a noise-free image, and noise, respectively.

Numerous approaches have been introduced to obtain noise-free images from the observed noisy ones; these approaches are generally categorized into two groups [14]: (1) model-based denoising methods [14,15,16,17,18], and (2) learning-based denoising methods [19,20,21,22].

Model-based denoising methods are considered as traditional techniques that use filters, such as median-type filter, Gaussian filter, and Gabor filter to remove noise. In efforts to remove impulse noise, Lin et al. [14] proposed a morphological mean filter that first detects the position and number of pixels without noise, and then uses these detected pixels to substitute neighboring noisy pixels in the image. Zhang et al. [15] introduced a new filter, named adaptive weighted mean (AWM). The main idea of this proposed filter is to reduce the error of detecting noise pixel candidates and replace them with suitable value computed using noise-free pixels. The proposed median filter [16] did not require any iteration to detect noises, but could directly identify low- and high-density salt-and-pepper noises as the lowest and highest values, respectively; it then utilized prior information to capture natural pixels and remove these noises. In the work [17], a hybrid filter combining fuzzy switching and median adaptive median filters was introduced for noise reduction in two stages. In the detection stage, the noise pixels are recognized by utilizing the histogram of the noisy images. Then, the detected noise pixels are removed whereas noise-free pixels are retained during the filtering stage to preserve the textures and details included in the original image. Based on the unsymmetrical trimmed median filter, Esakkirajan et al. [18] presented a modified decision method, in which the mean of all elements of a selected 3 × 3 window is used to replaced noise pixels for restoration of both color and gray images. In addition, various filtration algorithms [23], such as Wiener filtration, multistage Kalman filtration, and nonlinear Kalman filtration, which are commonly used for the formation of images containing characteristics close to those of real signals, can also be applied for denoising problems. However, although these approaches are simple to implement and achieve satisfying denoising performance in the presence of low-density noise, they are generally unsuitable for processing images featuring high-density noise.

Learning-based denoising methods consist of two main categories, including prior learning and deep learning. Between these methods, deep learning has shown outstanding results for noise removal in recent years. Xu et al. [19] recognized that simple distributions, such as uniform and Gaussian distributions are insufficient to describe the noise in real noisy images. They thus proposed a new method that could first learn the characteristics of real clean images and then use these characteristics to recover noise-free images from real noisy ones. The method in [20] removed noises and retrieved clean images by employing ant-colony optimization to generate candidate noise pixels for sparse approximation learning. Hieu et al. [21] introduced a lightweight denoising model using a convolution neural network (CNN) with only five deconvolutional and convolutional layers, which can be optimized in an end-to-end manner and achieves high running time on high-density noise images. In [22], the authors proposed a model based on a CNN, named DnCNNs, to denoise image with random noise levels. DnCNN accomplished the objective by modifying the VGG network [24] to extract features of the input samples, and engaging the batch normalization and residual learning for effective optimization. Unfortunately, while these methods yield high rates of success for denoising task, they require high computational costs and only work well when trained on a dataset with a massive number of image samples.

To reduce the dependence of the deep CNNs on a large training dataset, Ulyanov et al. [25] presented a deep image prior (DIP) approach, in which a handcrafted prior can be replaced with a randomly-initialized CNN for obtaining impressive results in common inverse tasks, such as blind restoration, super-resolution, and inpainting. Many research works based on DIP have been proposed [26,27,28], which introduce an additional regularization term to achieve robust performance. The method in [26] coped with the problem of denoising by combining both denoised and contrast-enhanced denoised results generated by DIP model. Cheng et al. [27] explained DIP from a Bayesian point of view and exploited a fully Bayesian posterior inference for better denoising result. In [28], DIP was adopted for image decomposition problems, such as image dehazing and watermark removal. The authors showed that stacking multiple DIPs and using the hybrid objective function allow the model to decompose an image into the desired layers.

Inspired by DIP, in this paper, we propose a novel method to address the problems of noise removal, which can reach remarkable denoising performance without necessitating a large training dataset. Our proposed model is comprised of three modules: (1) a DIP-based module, (2) an image fusion (IF) module, and (3) a progressive refinement (PR) module, as displayed in Figure 1. In the proposed model, first, the DIP-based module applies DIP to learn contrast enhancement and noise removal, simultaneously. Next, the IF module based on Laplacian pyramid (LP) decomposition, is used to avoid color shift and noise amplification during image production. Finally, the summed-area tables (SATs) are employed in the PR module to enhance edge information for acquiring high-quality output noise-free images of our proposed model. The details of these three modules are further presented in Section 2.

In summary, three technical contributions of our work are listed below:Noise removal and contrast enhancement are simultaneously learned for effective performance improvement without requiring large amounts of training data.The color distortion of denoised images is prevented by applying LP decomposition.Edge information is enhanced to achieve high-quality output images by taking advantage of spatially correlated information obtained from SATs.

To prove the efficiency of the proposed method, we use images from the DIV2K dataset [29] and create noisy images with various noise levels for evaluation. Our approach improved the results of competitive methods on 10 randomly selected test images by up to 16.7% in terms of noise removal.

The rest of our paper is organized as follows. In Section 2, we present the proposed method, including the DIP-based module, the IF module, and the PR module, in detail. Quantitative and qualitative evaluations, as well as a discussion of our findings, are provided in Section 3. Finally, we conclude our work in Section 4.

## 2. Proposed Method

This section presents a novel image denoising approach that could effectively remove noise and enhance edge information for image quality improvement. The proposed method is composed of three modules, namely, (1) a DIP-based module, (2) an IF module, and (3) a PR module, as illustrated in Figure 1. The details of each module are described in the following subsections.

### 2.1. DIP-Based Module

As mentioned in Section 1, numerous CNN-based noise removal models have been introduced to boost the denoising performance. However, these methods must be trained on a large dataset to learn better the features of images for the noise removal. In addition, the work [30] showed that although the deep CNN model learns well on a massive number of samples, it might also over fit the samples when labels are randomiized.

By contrast, not all sample priors need to be learned from the data, DIP [25] represents a self-supervised learning model based on a nested U-net with a randomly initialized input tensor trained on a noisy image for denoising; this model obtains impressive results compared with state-of-the-art denoising approaches. Thus, to enhance the denoised results, we first introduced DIP-based module to generate two different denoised images $I_{d}$ and $I_{ed}$. Our DIP-based module employs an encoder–decoder U-net architecture as a backbone network and relies on two target images, including a noisy image $I_{n}$ and an enhanced noisy image $I_{en}$, to supervise the production of $I_{d}$ and $I_{ed}$, respectively. Here, $I_{ed}$ is acquired by applying the contrast enhancement method [31] to $I_{n}$.

The architecture of the DIP-based module is displayed in Figure 2, where the input *z* and the output image have the same spatial resolution. The LeakyReLU [32] is used as a non-linearity. The strides executed within convolutional layers are applied for downsampling, while the bilinear interpolation and nearest neighbor techniques are utilized for upsampling. In all convolutional layers, except for the inversion of feature, zero padding is replaced with reflection padding. The detailed architecture of the DIP-based module is depicted in Table 1.

For training the DIP-based module, the objective function is described as follows.
(2)$$L_{dip}\;=\;E\left[||F\left(z,I_{n},\theta \right)\;-\;I_{n}{\left|\right|}^{2}\right]\;+\;E\left[||F\left(z,I_{en},\theta \right)\;-\;I_{en}{\left|\right|}^{2}\right],$$
where F(.) represents the DIP-based module, θ are parameters of F(.), and $F(z,I_{n},\theta )$ and $F(z,I_{en},\theta )$ represent the generated images $I_{d}$ and $I_{ed}$, respectively.

### 2.2. Image Fusion Module

Direct application of contrast enhancement to the noise removal stage may cause noise amplification and color shifts, resulting in the introduction of many visual artifacts [26]. To address this problem, we introduce an IF module to combine two denoised images $I_{d}$ and $I_{ed}$ generated by the DIP-based module.

LP was introduced by Burt et al. [33] to represent a compact image. The basic concept of the LP includes the following steps: (1) a lowpass filter *w*, such as the Gaussian filter, is used to transform the original image $I_{o}$ and downsample it by two to produce a decimated lowpass version; (2) the resulting image is upsampled by padding zeros between each column and row and convolving it with the filter *w* to interpolate the missing values and create an expanded image $I_{e}$. Subsequently, a detailed image $I_{di}$ is created by subtracting $I_{e}$ pixel by pixel from $I_{o}$. Based on the LP, many image fusion methods have been studied and acquired impressive results [34].

Inspired by the works in [33,34], our IF module employs LP decomposition to combine images effectively and prevent the image from shifting color and amplifying noise. Both denoised images $I_{d}$ and $I_{ed}$ are fed in the IF module and performed by the LP decomposition, denoted as $PL{\left(I_{d}\right)}^{n}$ and $PL{\left(I_{ed}\right)}^{n}$. The output image of the module is defined as follows:(3)$$I_{cd}\;=\;\alpha PL{\left(I_{d}\right)}^{n}\;+\;\left(1\;-\;\alpha \right)PL{\left(I_{ed}\right)}^{n},$$
where *n* = 1, 2,…, *l* denotes the *n*-th level in the pyramid, *l* denotes the number of pyramid levels, α is a hyperparameter, controlling the relative of two the LP decompositions. In our experiments, *l* is set to 5, and α is set to 0.7 to achieve the best results.

### 2.3. Progressive Refinement Module

After the combined image $I_{cd}$ is produced by the IF module, the edge information of this image is purified to improve the image quality. This objective is accomplished by utilizing our proposed PR module.

The PR module adopts SATs [35], which functions as a spatial filter and takes advantage of spatially correlated information for edge information enhancement. In the image $I_{cd}$, for each color component *c* with each pixel $I_{cd}\left(k,c\right)$ at position (x,y), the SAT calculates the summation of a patch as follows:(4)$$S\left(k,c\right)\;=\;\sum_{k^{\prime }\leq k}I_{cd}\left(k^{\prime },c\right)$$

We facilitate fast calculations by passing through the image with the SAT only once to achieve each pixel. Equation (4) is rewritten as follows:(5)$$S\left(k,c\right)\;=\;I_{cb}\left(k,c\right)\;+\;S\left(k+\lambda ,c\right)\;+\;S\left(k+\mu ,c\right)\;+\;S\left(k+\nu ,c\right),$$
where λ = (−1,0), μ = (0,−1), and ν = (−1,−1) present the direction offsets.

Through the SAT, the pixels that are unrepresentative of their surroundings in the image $I_{cd}$ are effectively eliminated. Thus, the proposed method could achieve enhanced edge information and generate high-quality images.

## 3. Experimental Results

All experimental results of our method for noise removal are summarized in this section. To conduct the experiments, we use the DIV2K dataset [29], randomly select 10 images and then resize these images to 512 × 512 for testing, as shown in Figure 3. Noisy images are obtained by applying additive white Gaussian noise (AWGN) [36] with amplitudes following a Gaussian distribution:(6)$$A\left(v\right)\;=\;\frac{1}{\sigma \sqrt{2\pi }}e^{-\frac{{}^{{\left(v-\mu \right)}^{2}}}{2\sigma^{2}}},$$
where σ and μ represent the noise standard deviation and mean value of the distribution, respectively. For zero-mean AWGN, μ is set to 0, and σ is a key parameter.

We create test noisy images by adding zero-mean AWGN with noise levels (σ) of 60, 70, and 80. Some examples of created noisy images are displayed in Figure 4. Since the number of runs necessary to obtain good denoised results are contingent upon the noise level, we run 1500, 1000, and 980 epochs on average to acquire our denoised results. For comparison, competitive denoising methods are used, including DIP [25], combination model of contrast enhancement method CLAHE [31] and DIP, denoted as CLAHE-DIP, and the method of Cheng et al. [26], denoted as DIPIF.

For quantitative assessment, we utilize a full-reference metric that exploits deep features to evaluate a perceptual similarity between a denoised image and a ground-truth image, called the Learned Perceptual Image Patch Similarity (LPIPS) metric [37]. Here, smaller LPIPS values indicate greater perceptual similarity of the compared images.

### 3.1. Ablation Study

To investigate the impact of some important design factors and select the best ones for our model, we conduct experiments using many different settings on a "chicken" image with 60%, 70%, and 80% noise corruptions, as shown in Figure 4. Four different structures are established to select the factors of the designed model: (1) model with DIP, $I_{d}$, $I_{ed}$, and the IF module, denoted as DIP-IF, (2) model with DIP, $I_{d}$, and the IF and PR modules, denoted as DIP-IF1-PR, (3) model with DIP, $I_{ed}$, and the IF and PR modules, denoted as DIP-IF2-PR, and (4) model with DIP, $I_{d}$, $I_{ed}$, and the IF and PR modules, denoted as DIP-IF-PR.

Figure 5 reveals that the DIP-IF-PR model achieves an LPIPS score of 0.318, which is better than DIP-IF, DIP-IF1-PR, and DIP-IF2-PR models by 1.6%, 1.1%, and 2.3%, respectively, in terms of denoising. Therefore, all the components used to constitute the DIP-IF-PR model are adopted in our proposed model, as shown in Figure 1.

### 3.2. Quantitative Results

Table 2 and Table 3 list the average LPIPS values and LPIPS values, respectively, computed by the compared methods on test noisy images with three noise levels of σ = 60, σ = 70, and σ = 80. The best LPIPS results achieved by the denoising methods are shown in boldface. As can be seen, our method surpasses the other methods on all randomly selected test images. The average LPIPS scores of compared methods on test images with 60% noise corruption are presented in the second column of Table 2. In this case, our proposed method reaches an LPIPS score of 0.248, outperforming the DIP$\left(I_{n}\right)$, DIP$\left(I_{en}\right)$, CLAHE-DIP, and DIPIF methods by 1.8%, 5.5%, 8.4%, and 1.1% in noise reduction, respectively. The average LPIPS scores of compared methods on test images with 70% noise corruption are shown in the third column of Table 2. Here, our proposed method obtains an LPIPS score of 0.283, improving the DIP$\left(I_{n}\right)$, DIP$\left(I_{en}\right)$, CLAHE-DIP, and DIPIF methods by 3.4%, 9%, 14.1%, and 1.5%, respectively, in terms of noise reduction. The average LPIPS scores of the compared methods on test images with 80% noise corruption are listed in the fourth column of Table 2. Our proposed method achieves an LPIPS score of 0.326, surpassing DIP$\left(I_{n}\right)$, DIP$\left(I_{en}\right)$, CLAHE-DIP, and DIPIF methods by 4%, 9.9%, 16.7%, and 2.3%, respectively, in terms of noise reduction.

### 3.3. Qualitative Results

We visually display the denoised image results of our proposed method and the competitive methods for various images corrupted by noise with three levels σ = 60, σ = 70, and σ = 80.

The denoised results of our method and compared methods on a “jellyfish” image, “house” image, and “statue” image with 60%, 70%, and 80% noise corruption are depicted in Figure 6, Figure 7 and Figure 8, respectively. As can be observed, the denoised results of the competitive methods still contain visible noises, resulting in blurred edges, darkening effects, and visual artifacts. By simultaneously optimizing denoising and contrast enhancement, and applying an edge enhancement technique, our method produces fewer artifacts and clearer natural-looking denoised images.

## 4. Conclusions

In this paper, we propose to apply deep image prior, image fusion, and edge enhancement techniques for the noise removal task. To succeed in denoising, our proposed method is structured using three modules, namely, the DIP-based module for concurrently learning noise reduction and contrast enhancement, the IF module for combining images and counteracting color shift and noise amplification, and the PR module for enhancing edge information. The experimental results on randomly selected test images proved that our method is able to yield satisfactory denoised images. Quantitative and qualitative assessments showed that the proposed method outperforms the competitive methods in terms of noise removal and image reconstruction.

Although the DIP is successfully applied to our proposed method for noise removal, the use of images $I_{d}$ and $I_{ed}$ as the inputs of the IF module can limit the speed of the whole model. This problem could be addressed by using features from the DIP module instead of its output images. We leave this work for future research.

## Figures and Tables

**Figure 1 sensors-21-05391-f001:**
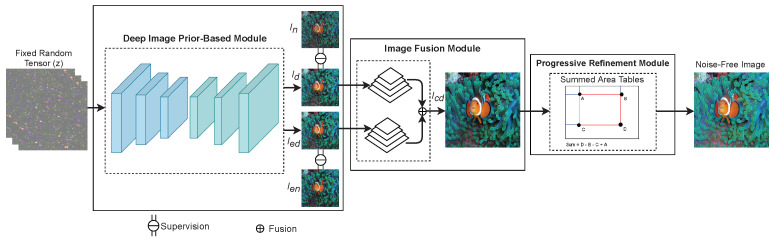
Flowchart of the proposed denoising method. $I_{n}$ is a noisy image, $I_{en}$ is an enhanced noisy image obtained by applying a contrast enhancement method to $I_{n}$. $I_{d}$ and $I_{ed}$ represent two denoised images that are supervised by $I_{n}$ and $I_{en}$, respectively.

**Figure 2 sensors-21-05391-f002:**
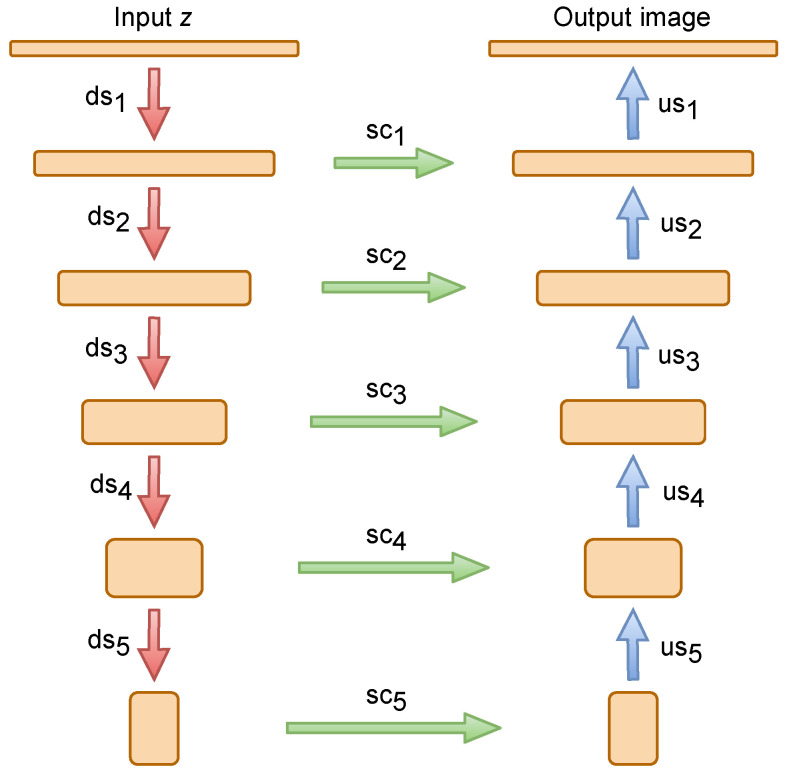
The architecture of the DIP-based module. Note that sc, us, and ds denote skip-connections, upsampling, and downsampling, respectively.

**Figure 3 sensors-21-05391-f003:**
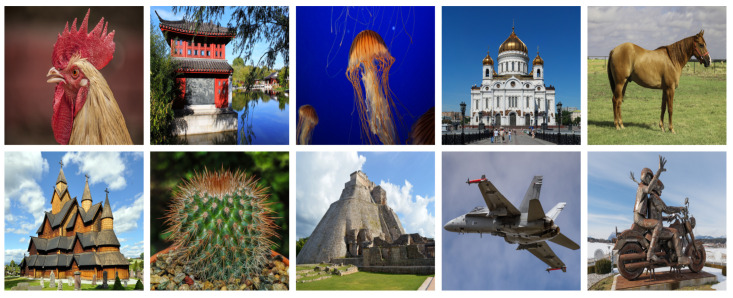
The images from the DIV2K dataset.

**Figure 4 sensors-21-05391-f004:**
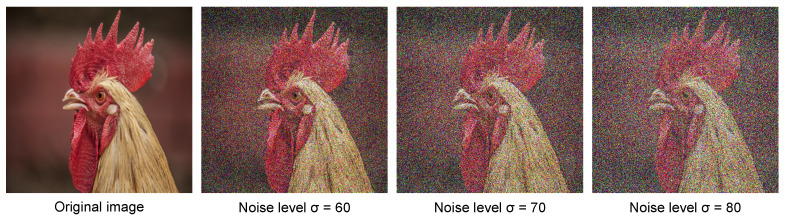
Example of “chicken” image with three noise levels.

**Figure 5 sensors-21-05391-f005:**
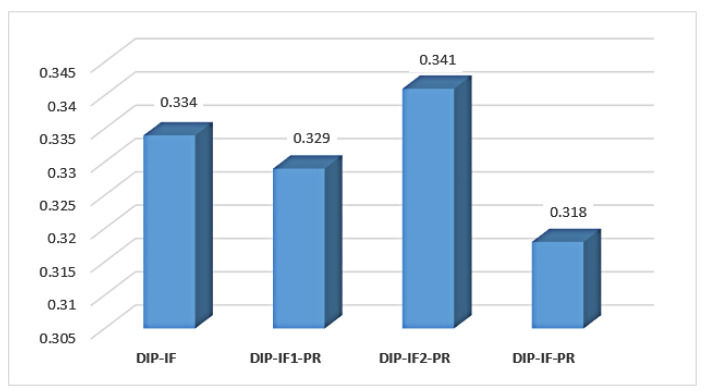
The average LPIPS scores of the models with different settings on a “chicken” image with 60%, 70%, and 80% noise corruptions. Note that a smaller value of LPIPS implies a better result in denoising.

**Figure 6 sensors-21-05391-f006:**
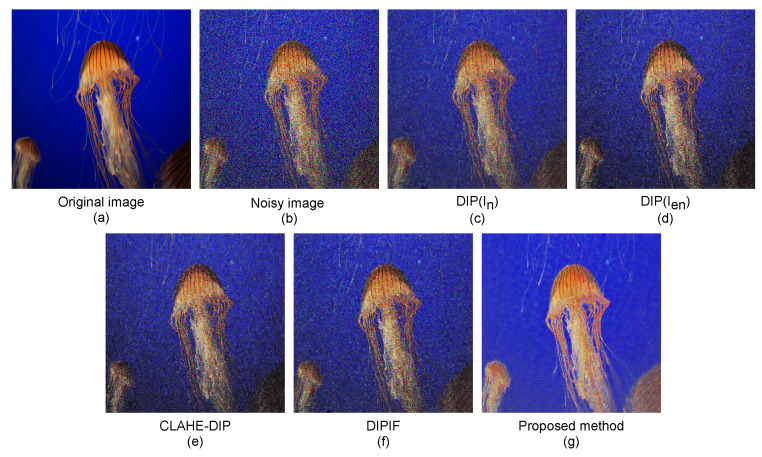
Noise-free images produced by our proposed method and competitive denoising methods on a “jellyfish” image with 60% noise corruption. (**a**) Clean image, (**b**) noisy image $I_{n}$, (**c**) image restored by the DIP method using the original noisy image $I_{n}$ for supervision, (**d**) image restored by the DIP method using enhanced noisy image $I_{en}$ for supervision, (**e**) image restored by the CLAHE-DIP method, (**f**) image restored by the DIPIF method, and (**g**) image restored by our proposed method.

**Figure 7 sensors-21-05391-f007:**
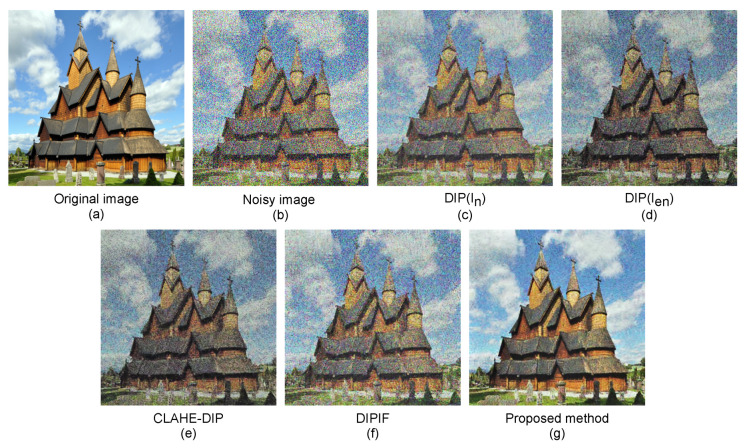
Noise-free images produced by our proposed method and competitive denoising methods on a “house” image with 70% noise corruption. (**a**) Clean image, (**b**) noisy image $I_{n}$, (**c**) image restored by the DIP method using the original noisy image $I_{n}$ for supervision, (**d**) image restored by the DIP method using enhanced noisy image $I_{en}$ for supervision, (**e**) image restored by the CLAHE-DIP method, (**f**) image restored by the DIPIF method, and (**g**) image restored by our proposed method.

**Figure 8 sensors-21-05391-f008:**
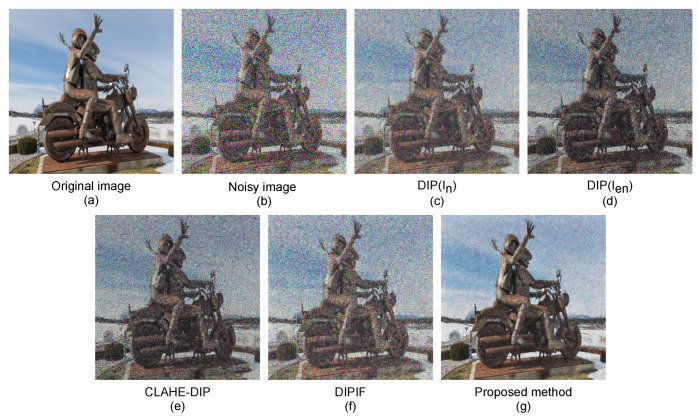
Noise-free images produced by our proposed method and competitive denoising methods on a “statue” image with 80% noise corruption. (**a**) Clean image, (**b**) noisy image $I_{n}$, (**c**) image restored by the DIP method using the original noisy image $I_{n}$ for supervision, (**d**) image restored by the DIP method using enhanced noisy image $I_{en}$ for supervision, (**e**) image restored by the CLAHE-DIP method, (**f**) image restored by the DIPIF method, and (**g**) image restored by our proposed method.

**Table 1 sensors-21-05391-t001:** The detailed architecture of the DIP-based module.

Type	Filter Size	Number of Filters
Downsampling	[3 × 3, 3 × 3, 3 × 3, 3 × 3, 3 × 3]	[128, 128, 128, 128, 128]
Upsampling	[3 × 3, 3 × 3, 3 × 3, 3 × 3, 3 × 3]	[128, 128, 128, 128, 128]
skip-connections	[1 × 1, 1 × 1, 1 × 1, 1 × 1, 1 × 1]	[4, 4, 4, 4, 4]

**Table 2 sensors-21-05391-t002:** The average LPIPS values of our proposed method and the competitive denoising methods on 10 randomly selected test images. Note that a smaller value of LPIPS implies a better result in denoising.

Method	Noise Level (σ)		
	60	70	80
DIP($I_{n}$) [25]	0.266	0.317	0.366
DIP($I_{en}$) [25]	0.303	0.373	0.425
CLAHE-DIP [25,31]	0.332	0.424	0.493
DIPIF [26]	0.259	0.298	0.349
Proposed Method	**0.248**	**0.283**	**0.326**

**Table 3 sensors-21-05391-t003:** The LPIPS values of our proposed method and the competitive denoising methods on 10 randomly selected test images. Note that a smaller value of LPIPS implies a better result in denoising.

Image	Chicken	Temple	Jellyfish	House 1	Horse	House 2	Cactus	Pyramid	Airplane	Statue
Noise Level	σ = 60									
DIP($I_{n}$) [25]	0.285	0.241	0.262	0.256	0.284	0.248	0.275	0.261	0.276	0.267
DIP($I_{en}$) [25]	0.339	0.269	0.305	0.278	0.327	0.275	0.313	0.296	0.319	0.310
CLAHE-DIP [25,31]	0.381	0.295	0.321	0.308	0.360	0.301	0.347	0.315	0.352	0.336
DIPIF [26]	0.273	0.238	0.259	0.251	0.271	0.246	0.267	0.256	0.269	0.262
Proposed Method	**0.265**	**0.225**	**0.247**	**0.237**	**0.259**	**0.236**	**0.256**	**0.245**	**0.251**	**0.254**
Noise Level	σ = 70									
DIP($I_{n}$) [25]	0.353	0.282	0.311	0.299	0.351	0.294	0.326	0.306	0.331	0.315
DIP($I_{en}$) [25]	0.408	0.318	0.379	0.354	0.399	0.349	0.384	0.375	0.386	0.381
CLAHE-DIP [25,31]	0.496	0.314	0.415	0.367	0.513	0.331	0.489	0.386	0.492	0.433
DIPIF [26]	0.327	0.255	0.298	0.279	0.333	0.272	0.307	0.289	0.317	0.302
Proposed Method	**0.316**	**0.244**	**0.277**	**0.26**	**0.304**	**0.281**	**0.292**	**0.274**	**0.293**	**0.286**
Noise Level	σ = 80									
DIP($I_{n}$) [25]	0.423	0.309	0.361	0.324	0.412	0.317	0.386	0.342	0.408	0.375
DIP($I_{en}$) [25]	0.496	0.347	0.412	0.388	0.485	0.363	0.455	0.398	0.478	0.425
CLAHE-DIP [25,31]	0.559	0.414	0.484	0.465	0.543	0.437	0.512	0.472	0.539	0.506
DIPIF [26]	0.404	0.288	0.345	0.303	0.391	0.294	0.382	0.325	0.393	0.364
Proposed Method	**0.374**	**0.270**	**0.323**	**0.298**	**0.362**	**0.283**	**0.348**	**0.312**	**0.355**	**0.330**

## Data Availability

Not applicable.

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
