# Peer review of "An Advanced Noise Reduction and Edge Enhancement Algorithm"

_sensors, 2021, doi:10.3390/s21165391_

Round 1
Reviewer 1 Report
This manuscript is based on a deep image prior (DIP) module, an image fusion (IF) module and a progressive refinement (PR) module, to process the image and effectively remove the noise. Simultaneous learning of the noise removal and contrast enhancement based on the DIP module greatly reduces the number of training data sets. Based on the IF module, the image's noise amplification and color shift are suppressed through Laplacian Pyramid (LP) decomposition. Based on the PR module, the summed-area tables (SAT) is used to achieve the purpose of enhancing the edge of the image and improve the image quality. The algorithm in this manuscript has made a certain contribution in noise reduction and edge enhancement. However, the following problems still exist:
(1)"DIP(Ien[24])" in Table 2 should be changed to "DIP(Ien)[24]". The calculation result of "8.5%" in line 195 should be "8.4%".
(2)The algorithm in this manuscript is very innovative, and the introduction of the algorithm is also very detailed, but in my opinion, the content at the end of section 1 and the content at the beginning of section 2 have some overlapping parts in the detailed description of the algorithm, and the corresponding parts may be deleted.
(3)The qualitative analysis of experimental results can be further supplemented, and some other indicators can be added to improve the persuasiveness of the optimization of the algorithm. (Such as PSNR index, SSIM index, FID index, etc.)
(4)For the three images listed in the quantitative analysis of the experimental results, the color of the three images is single, and images with various colors can be selected for quantitative analysis to enhance the proof of algorithm optimization.
Reviewer 2 Report
Authors demonstrate a new method of removing the noise of the image. The entire method consists of three major modules: the DIP module, the image fusion module, and an edge enhancement module. Both quantitative and qualitative characterizations are conducted and compared with other traditional denoising method. The effects indicate a fine noise removal. The manuscript is well-written and easy to follow. Some details and quantitative results should be given before considering the acceptance, in order to better evaluate the method.
- The quantitative results are given by evaluating the LPIPS metric. However, the individual effect of each module is unclear. It would be better to indicate the quantitative denoising effect after the image passing through each module to show the progressive effect. This would be significant especially the second and third modules are working on color correction and edge enhancement, respectively.
- It would be better to indicate the quantitative improvement in the introduction part, which can strengthen the contribution and differentiate the work from traditional methods.
- Some details should be given for the image process. For example, when passing through the DIP module, more details about the Id and Ied should be given, for example, would different contrast enhancement affect the final result?
- The application range should be specified. Can this method be applied to different distortion? For example, cases other than the gaussian noise. And how is the processing speed?
- For the noise reduction and edge enhancement, what is the smallest feature this method can reconstruct? It would be better to briefly introduce the potential error and solution of the method in the conclusion part.
Reviewer 3 Report
The article is devoted to the actual topic of image processing recorded by various sensors. The authors have developed a noise suppression method. The article proposes an image processing scheme that combines deep learning models and includes the following modules: Deep Image Prior-Based Module, Image Fusion Module, Progressive Refinement Module. The proposed method obtains 0.283 LPIPS score, improving known methods by 3-10% in average.
The main notes for the work are listed below:
1. Authors are encouraged to add to the overview of tasks on computer vision, tasks related to the preparation and cleaning of medical images, discussed in the article https://doi.org/10.3390/math9090967
2. In the review of model-based filtering algorithms, the authors should add an approach based on doubly stochastic random fields: and algorithms for filtering images with a complex structure from doi: 10.1134 / S1054661815010204.
3. A more detailed description of the noise and observation model is required. It is not clear what characteristics the noise has in the image. It is necessary to describe this model taking into account the subject of the journal, since it may be interference from the recording equipment (sensors).
4. It is necessary to investigate the LPIPS criterion for at least 10 different noise levels, averaging it over all images in order to obtain average comparative characteristics.
5. It is also recommended to give comparative characteristics of the execution time of the proposed algorithm and those with which the filtering results are compared.
6. The introduction should pay more attention to be consistent with the subject of the journal.
Round 2
Reviewer 3 Report
Good correction work!